# New Records of Sharpshooters (Hemiptera, Cicadellidae, Cicadellinae) in Citrus Orchards in Amazonas State, Brazil

**DOI:** 10.3390/insects15090649

**Published:** 2024-08-29

**Authors:** Paola Victoria Moreno Franco, Joyce Adriana Froza, Nathalia Hiluy Pecly, João Roberto Spotti Lopes, Jânia Lilia da Silva Bentes Lima, Agno Nonato Serrão Acioli

**Affiliations:** 1Programa de Pós-Graduação em Agronomia Tropical, Faculdade de Ciências Agrárias, Universidade Federal do Amazonas, Manaus 69067-005, AM, Brazil; jlbentes@ufam.edu.br (J.L.d.S.B.L.); acioli@ufam.edu.br (A.N.S.A.); 2Departamento de Entomologia e Acarologia, Escola Superior de Agricultura “Luiz de Queiroz”, Universidade de São Paulo, São Paulo 05508-220, SP, Brazil; jafroza@usp.br (J.A.F.); jrslopes@usp.br (J.R.S.L.); 3Departamento de Entomologia, Universidade Federal do Rio de Janeiro, Museu Nacional de Rio de Janeiro, Rio de Janeiro 20940-040, RJ, Brazil; nathalia.hiluy@gmail.com

**Keywords:** Amazonia, citrus variegated chlorosis, vector, *Xylella fastidiosa*

## Abstract

**Simple Summary:**

In our research conducted in seven citrus orchards in Amazonas State, Brazil, we recorded the presence of eight sharpshooter species. One of the identified species is known to transmit the bacterium *Xylella fastidiosa*, which is responsible for causing citrus variegated chlorosis (CVC). Among the eight species found, five were newly recorded in citrus plants within the State, and three were entirely new to the northern region. This research emphasizes the importance of mapping the locations of these insects to gain insights into their behavior. Doing so will enable the development of strategies to effectively manage the threat of CVC in local citrus crops.

**Abstract:**

This research study reports new records of eight species of Cicadellini (Hemiptera, Cicadellidae, Cicadellinae) across seven municipalities in Amazonas State, Brazil. Among these species, one is known as a vector of CVC (*Provancherana corniculata*). Additionally, five species are being reported for the first time in citrus for the State (*Erythrogonia sexguttata*, *Hortensia similis*, *Provancherana corniculata*, *Scopogonalia amazonensis*, and *Scoposcartula oculata*), and three species (*Diedrocephala variegata*, *Macugonalia moesta* and *Xyphon reticulatum*) are reported for the first time in the northern region. This research highlights the lack of information on sharpshooter occurrence in the Amazonas region, emphasizing the need for further investigations in this area.

## 1. Introduction

The Cicadellinae, also known as sharpshooters, is the third largest subfamily of Membracoidea (Hemiptera: Auchenorrhyncha) [1]. It includes two tribes: Proconiini Stål, 1869, which is limited to the New World and accounts for about 20% of the species, and Cicadellini Latreille, 1825, which accounts for around 80% of the species and is found in all zoogeographic regions, being particularly abundant in the Neotropical region (South America) [2,3,4]. This subfamily encompasses numerous species widely distributed across various biomes [5,6,7]. Cicadellinae species are polyphagous and can serve as vectors for *Xylella fastidiosa* Wells et al. 1987. The bacterium is responsible for citrus variegated chlorosis (CVC), is confined to xylem vessels, and causes diseases in various agricultural plants [8].

The Brazilian fauna currently comprises 625 species and 138 genera, with most species occurring in the Atlantic and the Amazon forests [9]. Most taxonomic work on descriptions comes from Amazonian native forests. However, there is limited information about the diversity of cicadelinaes due to inconsistencies in sample collection efforts across the region [10].

In this paper, we present the distribution of eight sharpshooter species in seven citrus orchards in Amazonas State, Brazil. We establish new occurrence records for Amazonas State and the northern region. Additionally, we reported the presence of *Provancherana corniculata* (Young, 1997), known as a vector of citrus variegated chlorosis (CVC). We aim to enhance our understanding of the geographical distribution of these species throughout the Amazon biome and across Brazil.

## 2. Methods

### 2.1. Study Area

The collections were conducted between March 2022 and April 2023 in sweet orange orchards situated in the municipalities of Beruri, Careiro, Iranduba, Itacoatiara, Manaus, Presidente Figueiredo, and Rio Preto da Eva in Amazonas State. The area under study is characterized as terra firme, and the climate in these areas is classified as type Af, tropical without a dry season [11]. The average annual temperature ranges from 25.9 to 27.7 °C, with a mean of 26.7 °C, and the annual precipitation in this region is 2420 mm. [11]. Most orange orchards vary in size and age, with monocultures dominating and surrounded by forests (Table 1; Figure 1).

### 2.2. Insect Sampling

Two types of collections were conducted in an orange orchard. In both cases, we captured sharpshooters using a sweep net with a diameter of 40 cm. For the ground-level collection, we chose 10 linear transects of 50 m each. Along these transects, we performed 80 sweeps in the undergrowth, which varied in height up to 50 cm (Figure 1). To collect from the canopy, 30 trees in the same rows were selected, and sweeps were carried out along the entire outer edge of each tree in the canopy.

### 2.3. Identification

The sharpshooters were identified by examining their morphological characteristics and comparing them to genus reviews, species descriptions [3,12,13,14,15,16,17,18,19,20,21], and digital images of Cicadellinae on the website “Sharpshooter Leafhoppers of the World” [22].

The male terminalia were prepared following the method outlined by Azevedo-Filho and Carvalho [23]. The abdomens of the specimens were removed and soaked in a 10% NaOH solution, then boiled for 15 min, rinsed with water, and transferred to glycerol for further dissection and photography [23]. Images of the adult specimens and their dissected terminalia structures were taken using a Nikon SMZ25 trinocular stereomicroscope. The terminalia parts were preserved in small vials with glycerin and properly labeled. The body length of all specimens used in this study was identified and preserved in 70% alcohol. They were deposited in the Entomology and Acarology Laboratory’s collection (LEA) at the Universidade Federal do Amazonas/UFAM-CEA.

### 2.4. Map Record

The map depicting the geographic record of sharpshooter species was created using QGIS-OSGeo4w-3.34-1 (http://qgis.org) (acceded on 1 July 2024). Additionally, the distribution records were compiled from a literature review, which included published articles on faunistic records and surveys of sharpshooters across various regions of Brazil.

## 3. Results

A total of 1922 sharpshooter individuals were collected from seven locations in the Amazonas State, representing eight species (Figure 2). *Scopogonalia amazonensis* Leal & Creão-Duarte, 2016 (seven individuals) was recorded for the second time in Amazonas (Leal et al., 2016), albeit with a new collection site (Figure 3). Similarly, *Macugonalia moesta* (Fabricius, 1803) (463 individuals), *Provancherana corniculata* (Young, 1977) (622 individuals), *Erythrogonia sexguttata* (Fabricius, 1803) (31 individuals), *Diedrocephala variegata* (Fabricius, 1775) (three individuals), and *Hortensia similis* (Walker, 1851) (785 individuals) were also recorded. Consequently, we proposed formalizing and expanding the geographic distribution of these species in the Amazonas region. Additionally, *Xyphon reticulatum* (Signoret, 1854) (nine individuals) and *Scoposcartula oculata* (Signoret, 1853) (two individuals) were recorded for the first time in the northern region and Amazonas State, respectively (Figure 3).

### 3.1. Diedrocephala variegata (Fabricius, 1775)

Figure 2, Figure 3A and Figure 4A

New record. BRAZIL—Amazonas. Beruri; 3°43′38″ S, 61°15′00″ W; 23 April 2022; P.V.M. Franco leg.; orange orchard, sweeping net; 1♀, UFAM-CEA 0001. Careiro; 3°29′21.0″ S, 60°08′59.0″ W; 20 April 2023; P.V.M. Franco leg.; orange orchard; sweeping net; 1♂, UFAM-CEA 0002.Presidente Figueiredo; 1°26′28.2″ S, 60°15′17.3″ W; 8 March 2023; P.V.M. Franco leg.; orange orchard, sweeping net; 1♀, UFAM-CEA 0003.

Previous records in Brazil. Bahia [13,24,25]; Espírito Santo [13,26]; Minas Gerais [13]; Paraná [13,27,28]; Mato Grosso [13,28]; Rio de Janeiro [13,29]; Rio Grande do Sul [13,28,30,31,32,33,34,35]; Santa Catarina [13,28,36,37]; São Paulo [13,38].

Identification. Specimens of *Diedrocephala variegata* were identified using Young’s key to genera and the species’ original description [3]. Additionally, morphological characteristics, including overall coloration and those of the male genitalia, were compared to those described by Azevedo-Filho et al. [17,20]. *Diedrocephala variegata* is generally brown to black in color. The crown and pronotum are dark brown with irregularly distributed white spots. A yellow–white stripe runs along the dorsal portion of the mesonotum, and a whiter stain, thickly bordered by black, is located near the apex of the clavus. The apical region of the forewings exhibits variations in spot patterns and colorations, including prominent red lines [13,39].

### 3.2. Erythrogonia sexguttata (Fabricius, 1803)

Figure 2, Figure 3B and Figure 4B

New record. BRAZIL—Amazonas. Beruri; 3°43′38″ S, 61°15′00″ W; 23 April2022; P.V.M. Franco leg.; orange orchard, sweeping net; 1♂, UFAM-CEA 0004.Iranduba; 3°12′15.6″ S, 60°13′39.4″ W; 1 April 2022; P.V.M. Franco leg.; orange orchard, sweeping net; 1♂, 4♀, UFAM-CEA 0005; 1 Febrary 2023; ibid; ibid; 1♀, UFAM-CEA 0006.Itacoatiara; 2°56′08.0″ S, 59°09′18.6″ W; 5 May 2022; P.V.M. Franco leg.; orange orchard, sweeping net; 5♂, 13♀, UFAM-CEA 0007; 24 August 2022; ibid; ibid; 1♂, 2♀, UFAM-CEA 0008.Presidente Figueiredo; 1°26′28.2″ S, 60°15′17.3″ W; 8 March 2023; P.V.M. Franco leg.; orange orchard, sweeping net; 1♀, UFAM-CEA 0009.Rio Preto da Eva; 2°43′00.7″ S, 59°26′40.3″ W; 25 October 2022; ibid; ibid; 2 ♀, UFAM-CEA 0010.

Previous records in Brazil. Bahia [19]; Goiás [19]; Mato Grosso [19]; Mato Grosso do Sul [19]; Pará [12,19]; Rio de Janeiro [29]; São Paulo [40,41].

Identification. *Erythrogonia sexguttata* was identified based on Medler’s [12] review of genera and the work of Carvalho and Mejdalani [19]. The visual characteristics of our specimens matched the diagnostic characteristics provided in these references. The crown is reddish brown with a pale yellow, bifurcated, triangular spot on the posterior margin. This spot continues onto the pronotum, taking an oval shape. The forewings are reddish brown with three large, pale yellow spots bordered by black. One spot is located in the middle of the clavus, adjacent to the mesonotum’s apex. Another spot lies on the median corium, adjacent to the costal margin. The third spot occupies the apical portion of the clavus and the adjacent area of the corium.

### 3.3. Hortensia similis (Walker, 1851)

Figure 2 and Figure 3C

New record. BRAZIL—Amazonas.Beruri; 3°43′38″ S, 61°15′00″ W; 23 April 2022; P.V.M. Franco leg.; orange orchard, sweeping net; 2♂, 6♀, UFAM-CEA 0011; 6 October 2022; ibid; ibid; 1♂, 3♀, UFAM-CEA 0012.Careiro; 3°29′21.0″ S, 60°08′59.0″ W; 12 August 2022; P.V.M. Franco leg.; orange orchard, sweeping net; 9♂, 11♀, UFAM-CEA 0013.Iranduba; 3°12′15.6″ S, 60°13′39.4″ W; 01 April 2022; P.V.M. Franco leg.; orange orchard, sweeping net; 86♂, 72♀, UFAM-CEA 0014; 1 February 2023; ibid; ibid; 35♂, 20♀, UFAM-CEA 0015.Itacoatiara; 2°56′08.0″ S, 59°09′18.6″ W; 5 May 2022; P.V.M. Franco leg.; orange orchard, sweeping net; 102♂, 82♀ UFAM-CEA 0016; 24 August 2022; ibid; ibid; 17♂, 5♀, UFAM-CEA 0017.Manaus; 2°52′44.6″ S, 60°04′39.9″ W; 11 March 2022; P.V.M. Franco leg.; orange orchard, sweeping net; 124♂, 96♀, UFAM-CEA 0018; 29 August 2022; ibid; ibid; 26♂, 32♀, UFAM-CEA 0019. Presidente Figueiredo; 1°26′28.2″ S, 60°15′17.3″ W; 6 December 2022; P.V.M. Franco leg.; orange orchard, sweeping net; 2♂, 2♀, UFAM-CEA 0020.Rio Preto da Eva; 2°43′00.7″ S, 59°26′40.3″ W; 18 April 2022; P.V.M. Franco leg.; orange orchard, sweeping net; 31♂, 21♀, UFAM-CEA 0021.

Previous records in Brazil: Bahia [24,25,28]; Espírito Santo [26]; Mato Grosso [28]; Para [42] Paraná [27,43,44]; Pernambuco [45]; Rio de Janeiro [29]; Rio Grande do Sul [28,30,31,32,33,34,35]; Roraima [46]; Santa Catarina [37,47]; São Paulo [38,40,48,49,50]; Sergipe [51].

Identification. We identified this species using Young’s key guide [3] to morphological characteristics and compared them to our specimens. Additionally, we consulted the illustrations provided by Azevedo-Filho et al. [17,20] for reference. *Hortensia similis* is characterized by a bright green coloration. The crown and anterior margin of the pronotum are light green, with a series of black linear marks and a central stripe. This stripe bifurcates anteriorly, forming the base of a triangle and containing two black spots at the line of the anterior mesonotum. The edges and posterior portion of the forewings are colorless, while the hindwings are brownish black. The clypeus is brownish yellow, and the anterior portion of the scutellum is yellowish with black marks. Notably, the clypeus is brownish yellow, but the chest and legs are yellow.

### 3.4. Macugonalia moesta (Fabricius, 1803)

Figure 2, Figure 3D and Figure 4C

New record. BRAZIL—Amazonas.Beruri; 3°43′38″ S, 61°15′00″ W; 23 April 2022; P.V.M. Franco leg.; orange orchard, sweeping net; 3♀, UFAM-CEA 0022.Careiro; 3°29′21.0″ S, 60°08′59.0″ W; 20 April 2023; P.V.M. Franco leg.; orange orchard, sweeping net; 1♂, UFAM-CEA 0023. Iranduba; 3°12′15.6″ S, 60°13′39.4″ W; 1 April 2022; P.V.M. Franco leg.; orange orchard, sweeping net; 225♂, 159♀, UFAM-CEA 0024; 1 February 2023; ibid; ibid; 1♀, UFAM-CEA 0025.Itacoatiara; 2°56′08.0″ S, 59°09′18.6″ W; 5 May 2022; P.V.M. Franco leg.; orange orchard, sweeping net; 15♂, 18♀, UFAM-CEA 0026; 24 August 2022; ibid; ibid; 5♂, 1♀, UFAM-CEA 0027.Manaus; 2°52′44.6″ S, 60°04′39.9″ W; 11 March 2022; P.V.M. Franco leg.; orange orchard, sweeping net; 1♂, 1♀ UFAM-CEA 0028; 29 August 2022; ibid; ibid; 4♂, 6♀, UFAM-CEA 0029.Presidente Figueiredo; 1°26′28.2″ S, 60°15′17.3″ W; 6 December 2022; P.V.M. Franco leg.; orange orchard, sweeping net; 12♂ 11♀, UFAM-CEA 0030.

Previous records in Brazil.

Identification. This species was identified using Young’s key guide [3], which relies on morphological characteristics, particularly the position of the aedeagus. *Macugonalia moesta* has a bright coloration. A bright blue lateral band covers most of the crown, extending to the apex of the margins of the compound eyes. The apical crown appears black and round in dorsal view. Additionally, two lateral bands adorn the pronotum, and a black band borders the anterior margin of the mesonotum. The forewings display a thick blue band adjacent to the posterior apex of the mesonotum. This band expands toward the apical portion of the clavus, where it merges to form a heart-shaped marking.

### 3.5. Provancherana corniculata (Young, 1977)

Figure 2, Figure 3E and Figure 4D

New record. BRAZIL—Amazonas. Beruri; 3°43′38″ S, 61°15′00″ W; 23 April 2022; P.V.M. Franco leg.; orange orchard, sweeping net; 5♂, 13♀, UFAM-CEA 0031; 6 May 2023; ibid; ibid 2♀, UFAM-CEA 0032.Careiro; 3°29′21.0″ S, 60°08′59.0″ W; 12.VIII.2022; P.V.M. Franco leg.; orange orchard, sweeping net; 94♂, 90♀, UFAM-CEA 0033.Iranduba; 3°12′15.6″ S, 60°13′39.4″ W; 1 April 2022; P.V.M. Franco leg.; orange orchard, sweeping net; 46♂, 28♀, UFAM-CEA 0034; 01 February 2023; ibid; ibid; 7♂, 5♀, UFAM-CEA 0035.Itacoatiara; 2°56′08.0″ S, 59°09′18.6″ W; 5 May 2022; P.V.M. Franco leg.; orange orchard, sweeping net; 4♂, UFAM-CEA 0036; 24 August 2022; ibid; ibid; 1♂, UFAM-CEA 0037.Manaus; 2°52′44.6″ S, 60°04′39.9″ W; 11 March 2022; P.V.M. Franco leg.; orange orchard, sweeping net; 70♂, 49♀, UFAM-CEA 0038; 29 August 2022; ibid; ibid; 81♂, 84♀, UFAM-CEA 0039.Presidente Figueiredo; 1°26′28.2″ S, 60°15′17.3″ W; 6 December 2022; P.V.M. Franco leg.; orange orchard, sweeping net; 2♀, UFAM-CEA 0040.Rio Preto da Eva; 2°43′00.7″ S, 59°26′40.3″ W; 18 April 2022; P.V.M. Franco leg.; orange orchard; sweeping net; 18♂, 12♀, UFAM-CEA 0041; 25 October 2022; ibid; ibid; 4♂, 7♀, UFAM-CEA 0042.

Previous records in Brazil. Bahia [28]; Mato Grosso [28] Pará [42]; Paraná [43]; Rio de Janeiro [29]; Rio Grande do Sul [28,31]; Roraima [46]; Santa Catarina [47]; São Paulo [28,38,40,41,48,49].

Identification. Species identification was based on the description provided by Young [3]. The crown is pale yellow, with two black circles encircling the ocelli, and a central black spot at the apex. In dorsal view, the apical crown appears rounded. The pronotum bears two transverse bands of a brownish color. The forewings are hyaline with prominent brown longitudinal veins [14].

### 3.6. Scopogonalia amazonensis Leal & Creão-Duarte, 2016

Figure 2, Figure 3F and Figure 4E

New record. BRAZIL—Amazonas. Careiro; 3°29′21.0″ S, 60°08′59.0″ W; 20 April 2023; P.V.M. Franco leg.; orange orchard, sweeping net; 6♂, 1♀, UFAM-CEA 0043.

Previous records in Brazil. Acre [21]; Amazonas, Itacoatiara [21].

Identification. We identified the species by comparing its morphological characteristics to the descriptions provided by Leal et al. [21]. These are green sharpshooters with translucent green forewings marked with scattered dark spots of varying sizes. The background color of the crown, along with the anterior third of the pronotum and mesonotum, is greenish yellow. The remaining dorsal surfaces are green. The crown bears a spot at the central apex of the anterior margin and possesses round green maculae surrounding the ocelli. These maculae are separated by a central black stripe. The pronotum has dark maculae located laterally on the posterior margin. The ventral body region is yellow, and the abdominal tergum is dark brown to black.

### 3.7. Scoposcartula oculata (Signoret, 1853)

Figure 2 and Figure 3G

New record. BRAZIL—Amazonas.Presidente Figueiredo; 1°26′28.2″ S, 60°15′17.3″ W; 18 October 2022; P.V.M. Franco leg.; orange orchard, sweeping net; 2♀, UFAM-CEA 0044.

Previous records in Brazil. Bahia, Goiás, Maranhão, Mato Grosso, Minas Gerais, Rondônia; São Paulo [16].

Identification. The identification of *Scoposcartula oculata* followed Leal et al. [16]. This species is a brown sharpshooter characterized by a punctate dorsal body surface. The crown has dark brown to black stripes along its lateral margins, which converge on the clypeus. The mesonotum has two dark bands, with the anterior band being wider than the posterior band. A dark brown to black macula adorns the scutellum, just anterior to the transverse sulcus. The forewings display a series of small white to yellow maculae scattered across the corium and clavus, positioned between the longitudinal veins from the base to the median region. Additionally, two large, yellow–white spots bordered by black are located at the apical portion of the clavus, adjacent to the costal margin.

### 3.8. Xyphon reticulatum (Signoret, 1854) Sensu Lato

Figure 2, Figure 3H and Figure 4F

New record. BRAZIL—Amazonas.Iranduba; 3°12′15.6″ S, 60°13′39.4″ W; 1 April 2022; P.V.M. Franco leg.; orange orchard, sweeping net; 1♂, UFAM-CEA 0045.Manaus; 2°52′44.6″ S, 60°04′39.9″ W; 11 March 2022; P.V.M. Franco leg.; orange orchard, sweeping net; 1♂, 1♀, UFAM-CEA 0046.Presidente Figueiredo; 1°26′28.2″ S, 60°15′17.3″ W; 6 December 2022; P.V.M. Franco leg.; orange orchard, sweeping net; 3♂, 3♀, UFAM-CEA 0047.

Previous records in Brazil. Rio de Janeiro [52]; São Paulo [53].

Identification. Species identification relied on the review of genera provided by Catanach et al. [18]. *Xyphon reticulatum* has a dark green coloration with variable marks on the crown. These marks are similar to those on the anterior pronotum. The central region of the pronotum is dark green, transitioning to yellow on the posterior portion. A well-defined median spot is present. The crown has a well-defined median spot and a dark brown medioapical macula. The median spot is encircled by cream pigment. Additional dark marks are visible, and the crown appears convex in lateral view. The pronotum exhibits dark green to brown circular marks with a white midline. The mesonotum is light green with two submedial spots. The forewings lack green pigmentation. Wings are green–brown. The anal veins are green, and the apex has few crossveins.

## 4. Discussion

Researchers have documented over 600 sharpshooter species and 100 genera among Brazilian fauna, with this number continuing to increase [54,55,56,57]. Out of this group, 15 vector species are known to transmit the bacterium to citrus, plums, and coffee trees. In southern Brazil, citrus cultivation has been extensively studied within the context of the CVC pathosystem [6,58,59]. However, there is limited information available on faunal surveys and species records of sharpshooters in Amazonian citrus orchards.

The Amazonas State ranks as the third-largest orange (*Citrus sinensis* (L.) Osbeck) producer in the northern region [60]. The cultivation area of orange orchards in Amazonas has rapidly expanded by over 100% in the past four years, increasing from 340 hectares in 2019 to 790 hectares in 2022 [61,62]. This significant growth has introduced various challenges that impact productivity, including phytosanitary issues. One particular concern is the presence of Cicadellinae species, which are known worldwide as important agricultural pests.

Eight sharpshooter species are formally recorded and distributed in seven citrus orchards in Amazonas State. Among these species, *Provancherana corniculata* is a vector of the *Xylella fastidiosa* bacteria within the CVC pathosystem [63]. The presence of this species poses a potential risk for the spread of CVC in the orange orchards of Amazonas, particularly due to its widespread occurrence in all the orange orchards.

The species *Diedrocephala variegata*, *Erythrogonia sexguttata*, *Hortensia similis*, and *Provancherana corniculata* are commonly found in orange groves, vineyards, weeds, and adjacent forests, particularly in the southern, southeastern, and central–western regions of Brazil where most of the faunal survey was carried out [30,37,38,41]. Most of these species were found in ground vegetation, and their presence in the orange tree canopy was minimal. *H. similis* and *P. corniculata* were more commonly found in weeds, with many individuals captured in this area [30,40], but they were also present in crops [41,48,50].

This study expands the known distribution of these sharpshooter species within the Amazonas State. *H. similis* and *P. corniculata* were recorded in orange orchards across seven municipalities. *Erythrogonia sexguttata* was found in five municipalities: Beruri, Iranduba, Itacoatiara, Presidente Figueiredo and Rio Preto da Eva. *Diedrocephala variegata* was found in three: Beruri, Careiro, and Presidente Figueiredo. These new records extend the known distribution range of these species by 33 km to 276 km. Notably, there are gaps exceeding 150 km between previously documented and present records.

*Macugonalia moesta* was not found in any of the faunal surveys analyzed. However, Young [3] indicated its occurrence in Brazil without specifying the biome. This study recorded its presence in six municipalities within the state: Beruri, Careiro, Iranduba, Itacoatiara, Manaus, and Presidente Figueiredo. It was also found in the Amazon area of Peru and Ecuador, on medicine, papaya, and avocado plants, respectively [64,65]. This widespread occurrence suggests that the species is well adapted to the Amazonian biome [52]. Notably, this study establishes the first record of *M. moesta* in the northern region of Brazil.

*Scopogonalia amazonensis* was originally described using forest material collected in the municipality of Itacoatiara [21]. In the present work, this species is recorded for the first time in orange orchards in the municipality of Careiro, which lies about 159 km west of Itacoatiara. This species differs from the original description due to the presence of certain morphological characteristics, such as scattered dark spots of varying sizes on the forewings [21].

This study represents the first recorded occurrence of *D. variegata*, *Scoposcartula oculata* and *Xyphon reticulatum* in the Amazonas State, marking a significant finding for these species.

*Diedrocephala variegata* is widely distributed across most Brazilian regions and can be found in different types of biome [13,24,26,28,37]. However, this study marks the first recorded occurrence of this species in the Amazonas State in the northern region. The taxonomic description of this species highlights morphological differences, such as the crown converging slightly on the clypeus, compared to the description of the species [16].

This study also established the first record of *X. reticulatum* in the northern region of the country. *Xyphon reticulatum*, previously known as *Carneocephala reticulatum* DeLong and Caldwell, 1937, was previously observed in invasive vegetation in citrus orchards in the southeast region [53]. In terms of identification, there are variations in morphological characteristics. According to Catanach [18], the *Xyphon reticulatum* complex includes *Xyphon sagittiferum* (Uhler, 1895), *X. dyeri* (Gibson, 1919), and *X. diductum* (Fowler, 1900) as synonyms, with morphological variations along latitudinal clines.

In conclusion, we formalized the recorded occurrence of sharpshooter species of the tribe Cicadellini in the Amazonas region, enhancing our understanding of entomofauna diversity. We also expanded the known geographical distribution within the Amazon biome, the northern region, and Brazil. These results underscore the current lack of information regarding the distribution of this group in Amazonas, and emphasize the urgent need for more taxonomic studies, particularly those describing new species. Increased collection efforts and studies on species distribution and diversity are crucial, given the economic significance of this group, which poses a threat to the spread of CVC in Amazonian citrus orchards.

## Figures and Tables

**Figure 1 insects-15-00649-f001:**
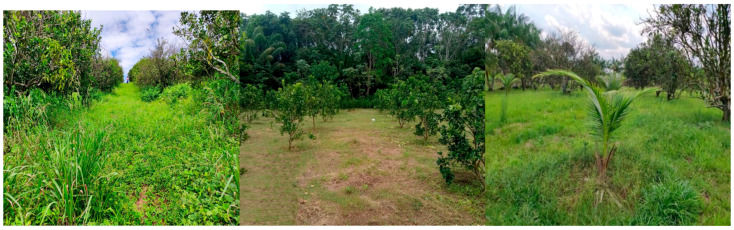
Three orange orchards are on display: one features tall vegetation, another is surrounded by forest, and the third includes a consortium of açaí and coconut plants.

**Figure 2 insects-15-00649-f002:**
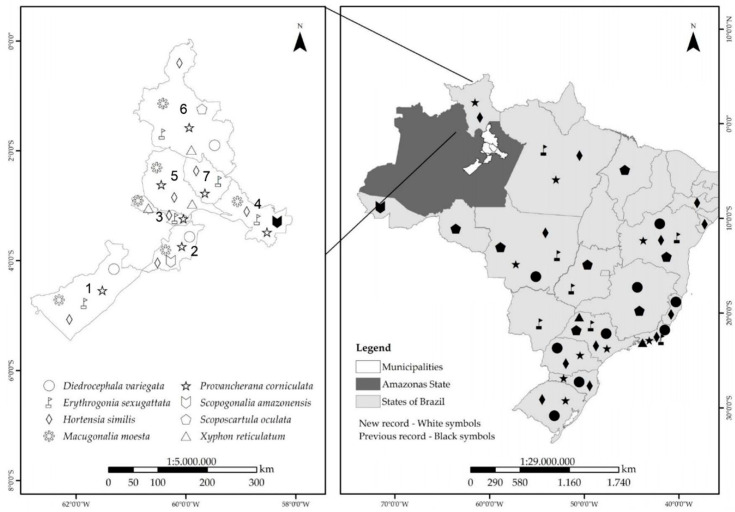
Geographic distribution of eight species of the tribe Cicadellini. Black symbols: previous records for Brazil; white symbols: new records for the Amazonas State. 1. Beruri. 2. Careiro. 3. Iranduba. 4. Itacoatiara. 5. Manaus. 6. Presidente Figueiredo. 7. Rio Preto da Eva.

**Figure 3 insects-15-00649-f003:**
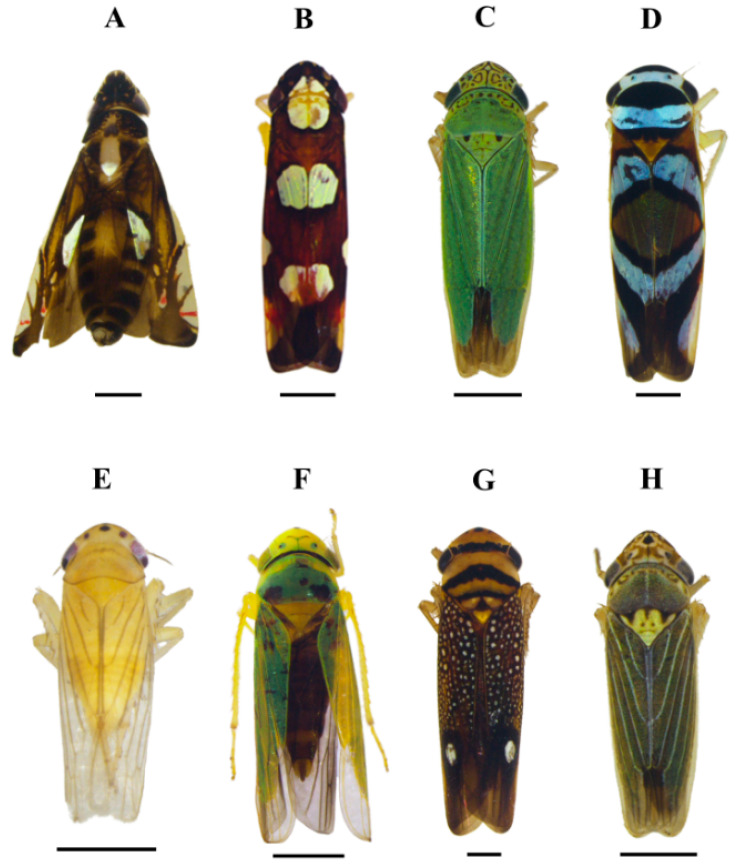
Sharpshooters collected with a sweep net in orange orchards in the Amazonas State, Brazil. Dorsal view. (**A**) *Diedrocephala variegata*. (**B**) *Erythrogonia sexguttata*. (**C**) *Hortensia similis* (**D**) *Macugonalia moesta* (**E**) *Provancherana corniculata*. (**F**) *Scopogonalia amazonensis*. (**G**) *Scoposcartula oculata*. (**H**) *Xyphon reticulatum*. Scale bar = 1.0 mm.

**Figure 4 insects-15-00649-f004:**
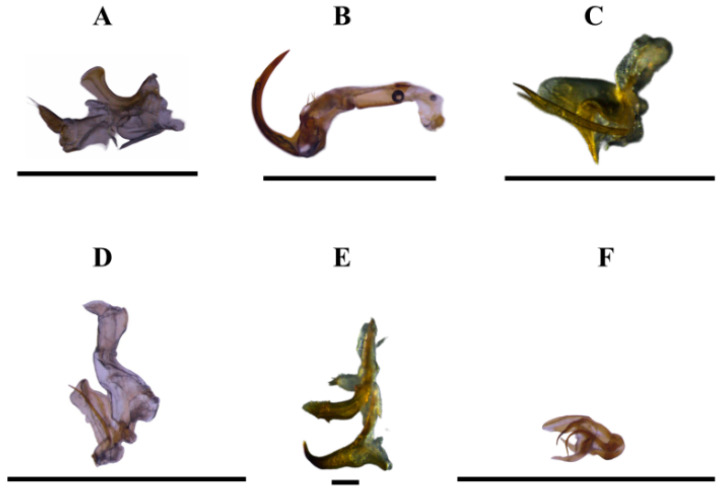
Sharpshooters collected with a sweep net in orange orchards in the Amazonas State, Brazil. Aedeagus lateral view. (**A**) *Diedrocephala variegata*. (**B**) *Erythrogonia sexguttata*. (**C**) *Macugonalia moesta*. (**D**) *Provancherana corniculata*. (**E**) *Scopogonalia amazonensis*. (**F**) *Xyphon reticulatum*. Scale bar = (**A**–**D**,**F**) = 1.0 mm; (**E**) = 0.10 mm.

**Table 1 insects-15-00649-t001:** Location of orange orchards where sharpshooters were collected with sweeping nets in Amazonas, Brazil.

Location/Orchard	Cardinal’s Point	Area of the Orchard (ha)	Year of the Orchard	Type of Cultivation	Vegetation Height	Surrounding Vegetation
Manaus	2°52′44.6″ S 60°04′39.9″ W	≅4.0	20	Monoculture	<50 cm	Forest/Orange
Rio Preto da Eva	2°43′00.7″ S 59°26′40.3″ W	≅12.0	10	Monoculture	<50 cm	Orange
Iranduba	3°12′15.6″ S 60°13′39.4″ W	≅5.0	10	Monoculture	<50 cm	Orange/others
Itacoatiara	2°56′08.0″ S 59°09′18.6″ W	≅2.5	25	Monoculture	<50 cm	Orange/others
Careiro *	3°29′21.0″ S 60°08′59.0″ W	≅2.0	20	Polyculture (Açaí & Coconut)	≥50 cm	Coffe/others
Beruri *	3°43′38″ S 61°15′00″ W	≅7.5	25	Monoculture	<50 cm	Forest/Orange
Presidente Figueiredo *	1°26′28.2″ S 60°15′17.3″ W	≅2.0	4	Consortium (Açaí)	≥50 cm	Forest/Orange

* = Information obtained from different orchards.

## Data Availability

All data that support the findings of this study are available in the main text.

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
