# Peer review of "New Records of Sharpshooters (Hemiptera, Cicadellidae, Cicadellinae) in Citrus Orchards in Amazonas State, Brazil"

_insects, 2024, doi:10.3390/insects15090649_

Round 1

Reviewer 1 Report

Comments and Suggestions for Authors This study reports the presence of 8 species of sharpshooters collected in Citrus orchards in the state of Amazonas, Brazil. The manuscript is well-written and the study is a useful one. Some of the names need to be changed to reflect current combinations. And the identifications of Hortensia similis should be changed to Hortensia sp. to reflect current knowledge of the group. An abstract by Prando & Takiya (2017) addressing Hortensia is appended to the attached pdf, and additional notes are included. The figures of the aedeagus are not of good quality, and I recommend that they not be included.

Author Response

New records of sharpshooters (Hemiptera, Cicadellidae, Cicadellinae) in citrus orchards in the Amazonas State, Brazil

Manuscript ID: insects-3052870

Response to Reviewer 1

In response to the recommendations and suggestions, we provide a point-by-point clarification and justification of the changes that have been made or not made following the article. Additionally, we want to emphasize that, based on the reviewers' recommendations and the journal's guidelines, we have excluded all citations from theses, dissertations, conference abstracts, and secondary sources as these have not been scientifically validated by peers.

Introduction

Page 1

Line 32: fourth (Deltocephalinae, Typhlocybinae, and Cixiinae are larger according to Bartlett et al.). 

Response: The text was modified based on the editor's suggestions.

"The Cicadellinae, also known as sharpshooters, is the third largest subfamily of Membracoidea (Hemiptera: Auchenorrhyncha)."

Methods

Page 3

Line 80: Which taxonomic works were consulted? e.g. Young, 1977.

Response: All of the taxonomic works used in the study were included in the text

Results

Page 3

Line 103: The current valid combination is Provancherana corniculata (Young, 1977).

Response: The suggestion was accepted, verified, and included in the text.

Line 104: Diedrocephala variegata (Fabricius, 1775) appears to be the valid name. See https://hoppers.speciesfile.org/otus/15368/overview. 

Response: The suggestion was accepted, verified, and included in the text.

Line 105: This should only be identified to the genus level, Hortensia sp. 

Response: The research by Prando & Takiya (2017) suggests that H. similis might consist a complex of similar species. However, it is important to note that this information comes from a meeting abstract and has not been published in a scientific journal.

Page 3

Line 111

Response: The suggestion was accepted, verified, and modified in the text. [Substitution of Diedrocephala bimaculata (Gmelin, 1789) for Diedrocephala variegata (Fabricius, 1775)].

Page 4

Line 153: Hortensia similis (Walker, 1851). 

Response: We maintain H. similis as it was explained in page 2, line 93.

Line 172: Modify this paragraph considering the work of Prando & Takiya (2017) noted above. 

Response: It is not accepted due to grey literature.

Page 5

Line 204. Plesiommata corniculata Young, 1977. 

Response: The suggestion was accepted, verified, and included in the text.

Page 7

Figure 3. These figures are of low quality and are not very useful. I recommend that they not be included. 

Response: We have decided to keep them as Reviewer 2 considers male terminalia to be the most relevant characters for correct specific identification

Reviewer 2 Report

Comments and Suggestions for Authors

 New records of sharpshooters (Hemiptera, Cicadellidae, Cicadellinae) in citrus orchards in the Amazonas State, Brazil

Authors: Paola Victoria Moreno Franco, Joyce Adriana Froza, Nathalia Hiluy Pecly, João Roberto Spotti Lopes, Jânia  Lilia da Silva Bentes Lima1 and Agno Nonato Serrão Acioli

This paper investigates the distribution of eight sharpshooter species in seven citrus orchards in Amazonas State, Brazil. The species studied are: Erythrogonia sexguttat a (Fabricius, 1803), Diedrocephala bimaculata (Gmelin, 1789), Hortensia similis (Walker, 1851), Macugonalia moesta (Fabricius, 1803), Plesiommata corniculata Young, 1977, Scopogonalia amazonensis Leal & Creão-Duarte, 2016, Scoposcartula oculata (Signoret, 1853) and Xyphon reticulatum (Signoret, 1854). Of these species, all except Scoposcartula oculata and Xyphon reticulatum were previously known in Amazonas state. This paper presents the first record of X. reticulatum in the northern region of Brazil. The authors provide species records based on their own work and existing literature, supplemented with photographs of habitus and genitalia.

The paper provides significant taxonomic data on sharpshooter species in citrus orchards in Amazonas, contributing to our understanding of their distribution. But the authors claim "new records" for species that are already known to occur in Amazonas state. Besides, the claims regarding their potential as vectors of Xylella fastidiosa are speculative and not supported by evidence within the paper or literature. To address these issues, the authors should reword the text to avoid making claims that are not supported by previous studies or their own data. The speculative nature of these claims needs to be acknowledged, and the term "potential vectors" should be used cautiously and relegated to the discussion section, not the abstract or main findings.

General comments

Misuse of the Term "New Records":

The authors claim "new records" for species that are already known to occur in Amazonas state. In taxonomic works, a "new record" refers to the first documented occurrence of a species in a specific geographical area where it was previously unreported. This includes:

    New Geographic Record: A species recorded in a new country, region, state, or island for the first time.

    New Habitat Record: A species discovered in a new type of habitat within its known geographic range.

Therefore, to avoid misleading use of the term "new records," the term "new" should be excluded from the title and the text, except for Scoposcartula oculata and Xyphon reticulatum, which are genuinely new records for the state of Amazonas.

Unclear New Habitat Records:

The paper does not clearly define new habitat records for the species. It states that the study offers valuable insights into the habits of these species but lacks tables or results detailing the habitats where these species were encountered. It is not clear if they were on citrus or on low vegetation within orchards. Furthermore, there is no comparison with previously published records of these species globally. For example, it is not clear if four of the eight sharpshooter species were recorded for the first time in citrus orchards (either in Brazil or globally).

Recommendations for Clarity on "New" Findings:

To be considered "new," the authors should explicitly state in the abstract, results, and discussion sections what is genuinely new. This includes specifying:

    New geographic records (if any).

    New habitat records (with clear documentation and comparisons).

    Any novel observations or findings related to the species' behaviors or interactions.

Speculative Nature of Vector Claims:

The authors sustain that the species of sharpshooters they study are potential vectors of Xylella fastidiosa, citing an unpublished PhD thesis. However, this thesis does not include any analysis of X. fastidiosa in the potential vectors, and therefore does not support their implications as vectors or even potential vectors. Besides, the authors say that in Brazil, 25 vector species transmit this bacterium to citrus, plums, coffee, and olive trees (Cornara et al., 2019; Müller et al., 2021; Carvalho et al., 2022; Froza, 2022). None of these works mention any of the species studied by the authors.

To substantiate the claim that these species are potential vectors, the paper or literature would need to include evidence such as:

    Molecular analysis confirming the presence of X. fastidiosa within these species.

    Behavioral studies demonstrating these species' interactions with infected plants.

    Epidemiological data linking these species to the spread of citrus variegated chlorosis.

Specific points of concern include:

    Unsupported Claims:

        "These insects are ecologically significant and can carry the bacterium Xylella fastidiosa, which causes citrus variegated chlorosis." There is no data to support this.

        "Providing essential data for understanding their ecological roles and associated disease risks." There is no data to support this.

        "By mapping their locations, our study offers valuable insights into their habits, aiding in developing strategies to manage the threat of X. fastidiosa in local citrus crops." There is no data to support this.

        "Focusing on their potential to carry the bacterium Xylella fastidiosa, which causes citrus variegated chlorosis (CVC) and significantly impacts crop yields." There is no information in the work that supports that these species are vectors of this disease. If they have been proved to be vectors, this should be stated in a table with all relevant literature for each species.

Contradictions:

    Line 42 states that citrus cultivation has been extensively studied within the context of the CVC pathosystem (Cornara et al., 2019; Carvalho et al., 2022). But line 45 says the opposite: "However, research on CVC disease and its vectors in this area is still in its early stages."

Title

Remove the word “new” and indicate the number of species studied. 

Abstract:

Include the species records and specify what is new in this work.

Line 30: Replace "ap" with "approximately."

Line 54: Clarify what is meant by taxa being misidentified.

Line 57: Specify that only some species are potential vectors of Xylella fastidiosa, as the current wording suggests that all species are vectors, which is not supported by evidence.

Lines 49-51: This paragraph refers to an unpublished work on population dynamics of leafhoppers, not on vectors. This should be removed or omitted. Since this is a taxonomic work, a section dedicated to previous studies on sharpshooters in Brazil is necessary, highlighting the number of known species, their endemic status, and current knowledge of this group. Reference all relevant works on sharpshooters in Brazil, noting how many are new or previously unknown for the Amazonas region.

Introduction

Line 30: Replace "ap" with "approximately."

Line 54: Clarify what is meant by taxa being misidentified.

Line 57: Specify that only some species are potential vectors of Xylella fastidiosa, as the current wording suggests that all species are vectors, which is not supported by evidence.

Lines 49-51: This paragraph refers to an unpublished work on population dynamics of leafhoppers, not on vectors. This should be removed or omitted. Since this is a taxonomic work, a section dedicated to previous studies on sharpshooters in Brazil is necessary, highlighting the number of known species, their endemic status, and current knowledge of this group. Reference all relevant works on sharpshooters in Brazil, noting how many are new or previously unknown for the Amazonas region.

Methods

Line 72: Specify which reviews and other literature were used for identification.

Line 78: The term "accurately identified" is redundant in a taxonomic work. Species are either identified or not.

Line 80: Describe how the species were photographed (e.g., in alcohol, dried) and how genital structures were prepared and preserved.

Line 81: Use "records" instead of "distribution," as the maps show species records, not true distributions.

Sampling Details: Include sampling dates, a map showing all sampling locations with states, municipalities, location names, and altitudes. Given the large size of Amazonas state, specify the most inclusive administrative areas.

Sampling Sites Description: Describe the sampling sites and include photos of the localities, biomes, and vegetation types. This is crucial even for species in agricultural areas.

  • example: Avivar-Lozano et al. (2024). "Three leafhoppers newly recorded from the European mainland (Hemiptera: Auchenorrhyncha: Cicadellidae), with notes on their habitats." Zootaxa, 5432(1), 1-18.

Results

  • Habitat and Vegetation: Address the type of habitats and vegetation where each species was studied. Provide an overview of these habitats.
  • Table: Include a table comparing species found in citrus and orange orchards, biomes  with other information from literature.
  • Nomenclature: Ensure proper taxonomic nomenclature. Author names should not be italicized. Include synonyms and avoid using "new records."
  • Species Records: Under each species name, cite the works, records name, and page.
  • Line 87: Avoid assuming species have expanded their geographic distribution. It is more likely they were always present but previously unreported.
  • Photos: Improve the resolution of photos and make genitalia figures more informative. If this is not possible, consider removing them.
  • Identification Details: Include the size of males and females. Compare genitalia with figures from published works, indicating the figures, pages, and commenting on any differences.

Discussion

Line 320: Correct the statement about the first record of M. moesta in Brazil. It was previously reported (see Line 194).

Focus: Concentrate the discussion on taxonomy and species distribution, as this is the main focus of the work.

References

Minimize the number of unpublished theses cited. Where possible, use published sources.

Comments on the Quality of English Language

no comments

Author Response

New records of sharpshooters (Hemiptera, Cicadellidae, Cicadellinae) in citrus orchards in the Amazonas State, Brazil

Manuscript ID: insects-3052870

Response to Reviewer 2

To follow the recommendations and suggestions, we provide a point-by-point clarification and justification of the changes that have been made or not made following the article. Also, we would like to emphasize that, based on the reviewers' recommendations and the journal's guidelines, we have excluded all citations from theses, dissertations, conference abstracts, and secondary sources as these have not been scientifically validated by peers.

General Comments 

Improper use of the term "New Records":

The authors claim "new records" for species whose occurrence is already known in Amazonas. In taxonomic works, a "new record" refers to the first documented occurrence of a species in a specific geographical area where it was not previously reported. To avoid misleading use of the term "new records," the term "new" should be excluded from the title and the text, except for Scoposcartula oculata and Xyphon reticulatum, which are genuinely new records for Amazonas state. 

Response: In Feitoza's Thesis (2017), the occurrence of the following species was recorded: D. variegata (bimaculata), E. sexguttata, H. similis, M. moesta, and P. corniculata. However, this information was not published in a scientific article. By referencing Feitoza's Thesis (2017), we are officially documenting the occurrence of these five species in the Amazonas state for the first time. If we disregard Feitoza's Thesis (2017), this would the first record for these five species. After considering the recommendations of reviewers, we have chosen not to include results from theses, dissertations, conference papers, and similar sources that have not undergone rigorous peer review. Therefore, this work does indeed represent the first recorded occurrence of the species: D. variegata (bimaculatta), E. sexguttata, H. similis, M. moesta, and P. corniculata, S. oculata, and X. reticulatum. With these clarifications, we have refined the literature by excluding all unpublished works from scientific journals and have retained the original proposed title.

Unclear new habitat records

The article does not clearly define new habitat records for the species. It states that the study provides valuable information on the habits of these species but lacks tables or results detailing the habitats where these species were found. It is not clear whether they were in citrus fruits or ground vegetation within the orchards. Furthermore, there is no comparison with previously published records of these species globally. For example, it is unclear if four of the eight leafhopper species were recorded for the first time in citrus orchards (in Brazil or worldwide). 

Response: Suggestion accepted.

Recommendations for clarity on "new" discoveries

To be considered "new," the authors must explicitly declare in the abstract, results, and discussion sections what is genuinely new. This includes specifying:

- New geographic records (if any).

- New habitat records (with clear documentation and comparisons).

- Any new observations or discoveries related to species behaviors or interactions.

Response: Suggestion accepted.

Reviewer 3 Report

Comments and Suggestions for Authors

This work involves a great effort in sampling and identifying the specimens. These identifications can be achieved by observing coloration characters, although the morphological characters, especially of the terminalia of males, are the most relevant to obtain a correct specific identification.

These and other species of shapshooters are important vectors of the bacteria Xylella fastidiosa in citrus, as in other crops, causing significant economic losses. The discovery of these species in citrus regions of northern Brazil is an important record when implementing appropriate management strategies to prevent the spread of CVC.

Please review the following comments:

page 3, line 100: no italics. please, review the species below

page 7, line 296: the number is 790?

Author Response

Response to Reviewer 3

To follow the recommendations and suggestions, we provide a point-by-point clarification and justification of the changes that have been made or not made following the article. Also, we would like to emphasize that, based on the reviewers' recommendations and the journal's guidelines, we have excluded all citations from theses, dissertations, conference abstracts, and secondary sources as these have not been scientifically validated by peers.

Page 3

Line 100: Without italics: 

Response: The suggestion was accepted and modified in the text

Page 7

Line 296: Is the number 790?

Response: The observation has been rectified in the text.